# Unexpected Method of High-Viscosity Shear Thickening Fluids Based on Polypropylene Glycols Development via Thermal Treatment

**DOI:** 10.3390/ma15175818

**Published:** 2022-08-24

**Authors:** Mariusz Tryznowski, Tomasz Gołofit, Selim Gürgen, Patrycja Kręcisz, Marcin Chmielewski

**Affiliations:** 1Faculty of Mechanical and Industrial Engineering, Warsaw University of Technology, Narbutta 85, 02-524 Warsaw, Poland; 2Faculty of Chemistry, Warsaw University of Technology, Noakowskiego 3, 00-664 Warsaw, Poland; 3Department of Aeronautical Engineering, Eskişehir Osmangazi University, Eskişehir 26040, Turkey; 4Faculty of Material Engineering, Warsaw University of Technology, Wołoska 141, 02-507 Warsaw, Poland; 5Institute of Microelectronics and Photonics, Łukasiewicz Research Network, Lotników 32/46, 02-668 Warsaw, Poland; 6National Centre for Nuclear Research, Materials Research Lab, Świerk, 05-400 Otwock, Poland

**Keywords:** shear thickening fluids, viscosity, thermal treatment, composite

## Abstract

This study aimed to analyze the influence of the thermal treatment of shear thickening fluids, STFs, on their viscosity. For this purpose, shear thickening fluids based on polypropylene glycols PPG400 and PPG1000 and Aerosil^®^200 were developed. The shear thickening behavior of obtained fluids was confirmed by using a parallel-plate rheometer. Next, thermogravimetric (TG) analyses were used to characterized thermal stability and weight loss of the STFs at a constant temperature. Finally, the thermal treatment of the STFs obtained was provided using the apparatus developed for this purpose. The received STFs exhibited a very high maximum viscosity up to 15 kPa. The rheology of the STFs measured after thermal treatment indicated that the proposed method allowed the development of STFs with a very high maximum viscosity. The maximum viscosity of the STFs increased twofold when thermal treatment of the STFs at elevated temperature for 210 min was performed. TG confirmed the convergence of the weight loss in the apparatus. Our results show that controlling the thermal treatment of STFs allows STFs to be obtained with high viscosity and a dilatation jump of the STFs by degradation of the liquid matrix.

## 1. Introduction

Shear thickening fluids, STFs, are intelligent materials that are characterized by the increase in their viscosity with either an increased shear rate or applied stress [1], known as a dilatation jump. This phenomenon can be observed as a transition from a liquid into a solid. Due to their properties, STFs can find several engineering and industrial applications. The most common application of STFs is their usage in armor protection. STFs are used to impregnate aramid fabrics to improve the protection of soft body armor [1,2,3,4]. Starch-based STFs show shear thickening properties suitable for soft body armor [5]. Furthermore, the use of STFs in combination with aluminum panels has also been shown in the literature [6]. STFs can be applied in shock-absorber systems for the automotive industry [7,8] or as protection for screen devices [9]. Furthermore, STFs have been applied to promote vibration damping in cutting tools [10] and as a fast, low-cost method for smoothing various surfaces [11,12].

In these materials, the solid powder nanoparticles are dispersed in a liquid matrix or a carrier fluid, forming a ceramic-polymer highly concentrated colloidal suspension. STFs are mainly composed of various inorganic powders, such as SiO_2_ [13], TiO_2_ [14], or CaCO_3_ [14] and liquid polymers such as glycerin [15], poly(ethylene glycol) [1,16,17], or poly(propylene glycol) [18,19,20]. Instead of SiO_2_, carbon nanofillers can be used [20]. Furthermore, with the addition of carbonyl iron powder to STFs, a fluid with magnetorheological properties can be developed [7]. Instead of inorganic particles, various polymers can be used. Synthetic polymers such as polystyrene-ethyl acrylate particles as solids can be used for STF development [6,21,22].

STFs are mainly prepared by stepwise addition of solids in nanoparticle size to the liquid matrix. This process requires appropriate mixing, which is limited by the increasing viscosity of the fluid being created [23]. Thus, the procedure involves the use of powerful mixers, and the fluid takes several hours to prepare [24]. Liu et al. prepared SiO_2_-poly(ethylene glycols) STFs by dispersing the silica in the liquid poly(ethylene glycol) 200 using a ball mill at room temperature [7]. Unfortunately, this process takes a long time, and the mixing was performed for 24 h. Some authors report the development of STFs by stepwise addition of carrier fluid to powder silica [18]. Furthermore, Mahesh et al. report using mixing and sonification to develop STFs by adding solids to a liquid matrix [21,25]. Furthermore, an emulsion polymerization can be used for the development of STFs [6,21]. Some authors proposed a synthesis of STFs by mixing the polymer matrix and silica in the excess of a solvent, such as ethanol [26]. This procedure needs an evaporation process of the solvent. The development of liquid STFs with a high dilatation method is impossible due to the increase in viscosity and a problem with the proper mixing and dispersing of the solid content in the liquid polymer matrix. However, the shear thickening effect is not fully understood [27]. There are a few studies that discuss the problem of modeling of the shear thickening effect of STFs [28,29,30,31,32].

In this work, we show a path to obtain high-viscosity STFs. The STFs were obtained by the traditional method of stepwise addition of fumed silica into the liquid PPG. Next, the shear thickening behavior was confirmed by rheology measurement. The degradation of the STF was measured with the TG technique at a constant temperature (100 °C, 110 °C, 120 °C, 130 °C, and 140 °C). The TG test was needed to assess the degradation behavior of STF and it was done for the selected STF with the highest dilatation effect. Finally, the thermal treatment of the STFs was performed, and their viscosity was confirmed. The unexpected effect of thermal treatment was an increase in the STFs’ viscosity. We suggest that thermal treatment in a mild condition can be a method for the development of high-viscosity STFs. The novelty of this work is focused on showing an easy way to obtain STFs with a high dilatant jump by degradation of the liquid matrix.

## 2. Materials and Methods

### 2.1. Materials

Poly(propylene glycol)s PPG400 and PPG1000 (CAS 25322-69-4, Sigma-Aldrich, St. Louis, MO, USA) were used as a liquid phase and Aerosil^®^200 (CAS 112 945-52-5, Evonik Industries, Hanau-Wolfgang, Germany) hydrophilic-fumed silica as a solid phase for STF preparation, respectively. Table 1 shows the properties of the materials used for STF preparation.

### 2.2. STFs Preparation

Table 2 shows the STFs’ formulation. The STFs were developed by the method shown in our previous work [27]. The liquid PPG was placed in a 250 mL reactor equipped with a mechanical stirrer (R50D, Ingenieurbüro CAT, Ballrechten-Gottingen, Germany) and a stainless-steel propeller-mixing geometry. Next, the silica was added stepwise in minimal amounts to the mixture. To achieve high-viscosity STFs, the stirring speed (from 200 rpm) was gradually decreased to a very low mixing speed (down to 1 rpm) as the silica was added. Stirring at nearly 1 rpm for 14 days was required to obtain a very viscous liquid (STF1000-24).

### 2.3. Rheology Measurement

A rotational MCR 102 rheometer (Anton Paar Company, Graz, Austria) equipped with a plate-to-plate (top plate *φ* 20 mm; bottom plate *φ* 100 mm; spacing gap 0.7 mm geometry) that applied shear stress (up to 250 s^−1^) was used to determine the shear thickening response of fluids. The rheological measurements were performed prior to and after the thermal treatment of the STFs. The measurement was repeated twice with a new sample.

### 2.4. Thermogravimetric Analysis

Thermogravimetric analysis (TG) was carried out using TA Instruments SDT Q600 (New Castle, DE, USA) apparatus. The weight loss was measured at constant temperatures of 100 °C, 110 °C, 120 °C, 130 °C, and 140 °C as a function of time. For TG measurement purposes, a sample of approx. 10 mg was used.

### 2.5. Thermal Treatment Procedure

In order to develop high-viscosity STFs, a special apparatus was designed. The computer-aided design (CAD) scheme of the apparatus is displayed in Figure 1. The weight loss procedure is schematically shown in Figure 2.

An insulated chamber with a movable carousel was heated by five heaters (Finger Patron Heater, Selfa GE SA, Szczecin, Poland), with dimensions 10 × 60 mm, at 300 W each. The power of the heaters was regulated by a Eurotherm 7100A thyristor controller (Worthing, UK) triggered by a PLC under the control of the TwinCAT 3.1.4024.29 system (Beckhoff Automation GmbH & Co. KG, Verl, Germany). The amount of heat supplied to the chamber was regulated by a PID controller. After the temperature inside the chamber was stabilized, six vessels with the STFs for testing were introduced. After the top insulated cover was closed, the control algorithm responsible for the controlled degradation process was triggered. During the degradation process, the sample carousel was rotated clockwise through an angle of 60° every 120 s (see Figure 2a). This ensured that the temperature of all the degraded samples was homogeneous. The AM3012-0C41 servo motor (Beckhoff Automation GmbH & Co. KG, Verl, Germany) with an absolute encoder and a mechanical gear PLE40-M01-10 with a gear ratio of 1:10, powered by the Beckhoff AX5203 module, was responsible for the correct positioning of the carousel (Beckhoff Automation GmbH & Co. KG, Verl, Germany). The weight loss was measured by a P/N 83020552 strain gauge beam from an Ohaus MB25 moisture analyzer (Parsippany, NJ, USA) with an accuracy of 0.005 g/0.05%. The strain gauge beam with the pan was placed on a linear sliding table (Figure 2b). The YR-GZS90K-100 (Lishui City Yongrun Precision Machinery Co., Ltd., Lishui, China) was equipped with a 57HD4016-01 stepper motor (Dongguan Golden Motor Co., Ltd., Dongguan, China) powered by a Beckhoff EL7041-1000 module (Figure 2c). Due to the micro-vibrations during the taring and weighing process, the stepper motor was turned off.

Measurement data (vessel weight) were periodically saved by the PLC system to the SQLite database. The sample of approx. 1 g was placed in the vessels (50 mm diameter), and the sample was taken for rheology measurement after a specified period of time. The samples’ weight loss was controlled to ensure that the decomposition of the STF samples was similar to that during TGA measurement.

## 3. Results and Discussion

### 3.1. STFs Properties

The STFs were developed with poly(propylene glycol)s having a molecular weight of 400 g/mol (PPG400) and 1000 g/mol (PPG1000) and fumed silica Aerosil^®^200 with a particle size of ~12 nm [33] and a specific surface area of ~200 m^2^·g^−1^. Using PPGs as a carrier fluid instead of PEGs (polyethylene glycols) resulted in higher viscosities at lower shear rates [34]. Furthermore, the melting points of PEGs were much higher than PPGs, and the STFs based on PEGs were temperature-sensitive at low temperatures [35].

Figure 3 presents the viscosity-shear rate dependence of the developed STFs with various concentrations of fumed silica dispersed in poly(propylene)glycol. It can be seen that the developed STFs exhibit typical shear-thickening behavior. It is crucial to obtain STFs with very low viscosity at low shear rates and a very high dilatation jump (or viscosity jump). In other words, the developed STFs should be a liquid, not a paste, and exhibit a high maximum viscosity upon stress. As expected, the increasing silica content resulted in a more significant dilatant effect. Simultaneously, the oligomer’s higher molecular weight provided higher viscosity values. The STF1000-24 and STF1000-18 exhibited a dilatant effect with a maximum viscosity of 1539 Pa·s and 830 Pa·s. Furthermore, the STFs based on PPG400 exhibited approx. three times lower values of maximum viscosity: 501 and 269 Pa·s for the STF400-24 and STF400-18, respectively. Comparing the results of the viscosity of the STFs already reported in the literature, it can be seen that the viscosities and the solid content achieved by us were much higher. Fisher et al. developed STFs based on Aerosil^®^200 and PPG1000 with a solid content of up to 15% *w*/*w* [18]. Wierzbicki et al. reported STFs with dilatant effects around 200 Pa·s using PPG400 and fumed silica with a specific surface 200 m^2^·g^−1^ [36]. Arora et al. and Bajya et al. reported the development of STFs with a maximum viscosity reaching 170 Pa s [37,38]. Furthermore, using hydrophobic silica with the same particle size and specific surface area and PPG400 and PPG1000, even when introducing more content of solids, resulted in much lower viscosities (up to 62 Pa·s). Nevertheless, in our previous work, using the same methodology of STF development, we reported fluids with a maximum viscosity exceeding 3000 Pa·s [27].

### 3.2. Thermal Treatment

In this work, the thermal decomposition of the developed STF1000-24 was performed with TG (thermogravimetric analysis). This method was used to control weight loss changes and to confirm the influence of the temperature on the nature of the degradation. However, it was not possible to compare the TG degradation course and thermal treatment directly, because the sample mass was 100 times larger. Additionally, TG makes it possible to determine a thermal decomposition with a very small sample (approx. 10 mg) and rheological measurement requires much bigger samples (approx. 0.7 g). Hence, we developed an apparatus that allows the thermal degradation of materials in similar conditions to those in TG.

The TG degradation was performed at five various constant temperatures (100 °C, 110 °C, 120 °C, 130 °C, and 140 °C) for the STF1000-24 as shown in Appendix A. Figure 4 shows the TG curves of STF1000-24 at 140 °C. According to the data (see Table 3, Appendix A and Figure 4), three steps of thermal degradation could be identified for the STF1000-24 sample, independently of the degradation temperature. The weight loss was accompanied by an exothermic effect. These steps were characterized by various weight loss and heat effects. The first step (I) was related to a slight heat effect connected with a weight loss rate of 0.02% and 0.18% per min at 100 °C and 140 °C, respectively. It might have been attributed to the evaporation of low molecular mass glycols. As shown in Figure 4, no changes in STFs were observed in step I. The second one (II) was characterized by a faster weight loss rate: 0.05% and 0.37% per min at 100 °C and 140 °C, respectively, and it was accompanied by a bigger heat effect. We suspect that this step was related to the vaporization and burning of volatile compounds arising during the degradation of the poly(propylene glycol) molecules. Finally, in the third step (III), the weight loss speed was the highest, leading to the decomposition of the STFs. Step III was related to visible changes in the appearance of the material. The non-dispersed silica appeared on the surface of the fluid. The maximum heat effect of the decomposition was approx. 8 times higher at 140 °C than at 100 °C. Hence, the reaction was exothermic and due to the high maximum heat effect at higher temperatures, during thermal treatment or mixing at elevated temperatures, the mixture could have overheated, the heat could have accumulated, and accelerating decomposition was observed. Furthermore, carrying out the process at an elevated temperature might have led to an uncontrolled explosion of heat. Therefore, the process should be performed at lower temperatures on a large scale.

The samples of STFs were thermally treated at temperatures of 100 °C, 120 °C, and 140 °C with the apparatus shown in Figure 1. The control lines of weight loss are shown in Figure 5. The rotating carousel made it possible to ensure the homogeneous temperature of all the degraded samples. After a specified period of time, the samples of STFs were taken for viscosity measurement. According to the control lines of thermal treatment, the approx. 25% weight loss was observed after 210 min of thermal treatment.

Figure 6 shows the viscosity of the obtained STF1000-24 after thermal treatment for 210 min at 140 °C. The exposure of the fluids to elevated temperature caused the gradual degradation of the liquid matrix in the STFs, which could be observed as an increase in viscosity. The liquid matrix content decreased by approx. 25%, so the degradation was in the second step of degradation (see Figure 4). A higher reduction in the liquid matrix might have caused irreversible changes in the STFs and total fluid degradation. Hence, the approx. 200 min of thermal treatment at 140 °C seemed to be optimal, allowed to be with the degradation in step II, and prevented the total degradation of the STF and loss of the shear thickening properties of the STF. As shown in Figure 6, the viscosity of STF400-18 and STF1000-18 after thermal treatment increased twofold (blue and gray dashed lines), reaching maximum viscosity of 542 Pa·s and 1632 Pa·s, respectively. It can be seen that the maximum viscosity after thermal treatment had increased.

Last but not least, we also thermally treated the STF1000-24 at 140 °C. The process was carried out at a constant temperature until the desired mass was achieved: 15%, 20%, and finally 25% weight loss. The viscosities of the STF1000-24 samples after the specific thermal treatment course are presented in Appendix A. The thermal treatment of STF1000-24 with the degradation of 15% resulted in a maximum viscosity of 6 kPa·s. Further degradation of STF1000-24 revealed a fluid with maximum viscosity at 8.5 kPa·s (degradation of 20% of the matrix) and 14 kPa·s (25% matrix degradation), respectively.

## 4. Conclusions

In this work, we developed four types of STFs based on poly(propylene glycol)s and fumed silica with high solid content (18% and 24% *w*/*w*). The developed STFs exhibited high-viscosity properties with a maximum viscosity level at 1539 Pa·s for STFs based on PPG1000 and fumed silica with 24% solid content. The STFs were thermally treated, which allowed a twofold increase in the viscosity properties of the obtained STFs.

We showed that controlled thermal treatment in mild conditions of low viscosity STFs can be used as a tool to obtain STFs characterized by high viscosity and high solid content. Our results reveal the possibility of using thermal treatment at elevated temperature to increase the viscosity of STFs as a novel method of high-viscosity STF preparation. This is a breakthrough paper in the topic of STFs because it shows the importance of controlling the temperature during the development of the fluids. Overheating the STFs during development can lead to complete degradation, but controlled heating can significantly increase the viscosity without mixing problems.

## Figures and Tables

**Figure 1 materials-15-05818-f001:**
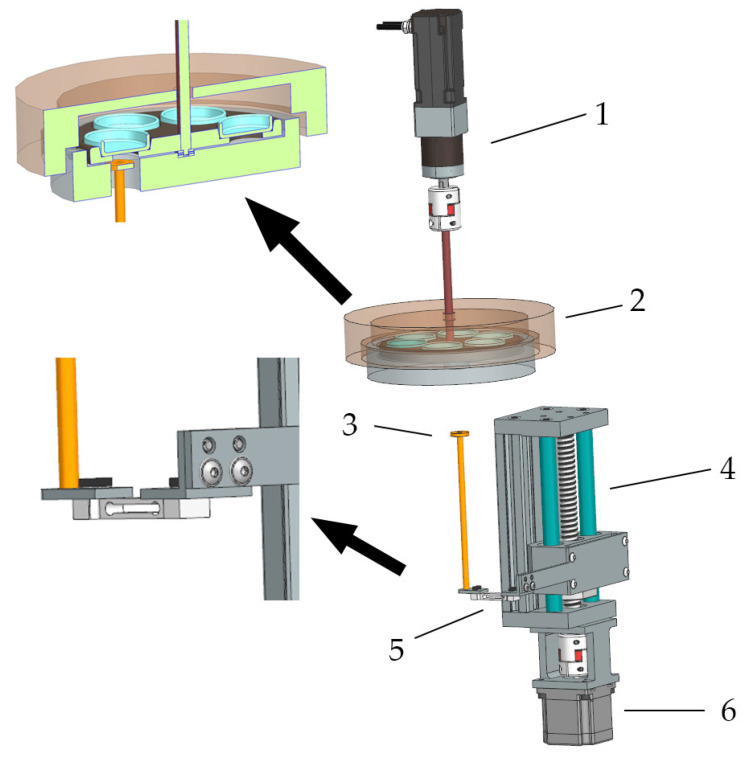
Apparatus scheme for degradation process: 1—servo motor with gearbox and encoder; 2—vessel carousel with heating; 3—beam balance; 4—linear table; 5—load cell; 6—stepper motor.

**Figure 2 materials-15-05818-f002:**
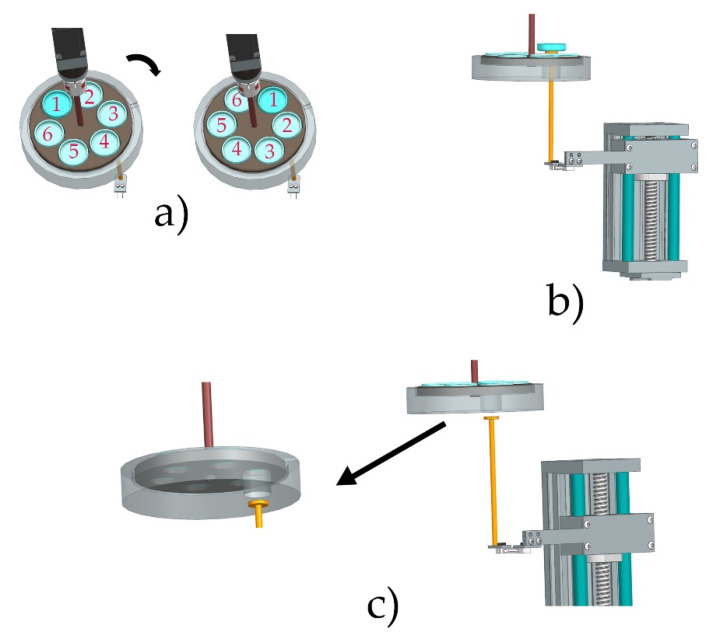
The weight loss measurement procedure: (**a**) clockwise rotation through an angle of 60°; (**b**) weighing of the vessel; (**c**) balance arm retraction and balance adjustment.

**Figure 3 materials-15-05818-f003:**
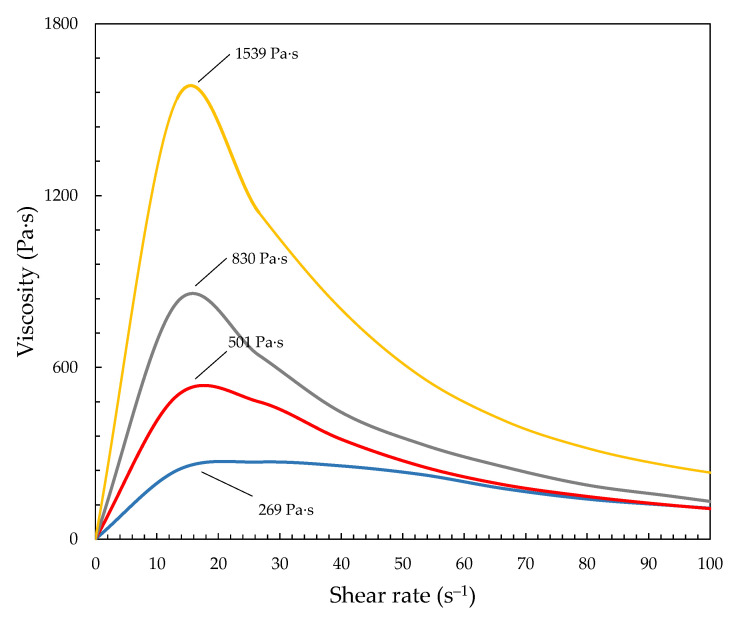
Viscosity vs. shear rate for developed STFs: yellow line—STF1000-24; gray line—STF1000-18; red line—STF400-24; blue line—STF400-18.

**Figure 4 materials-15-05818-f004:**
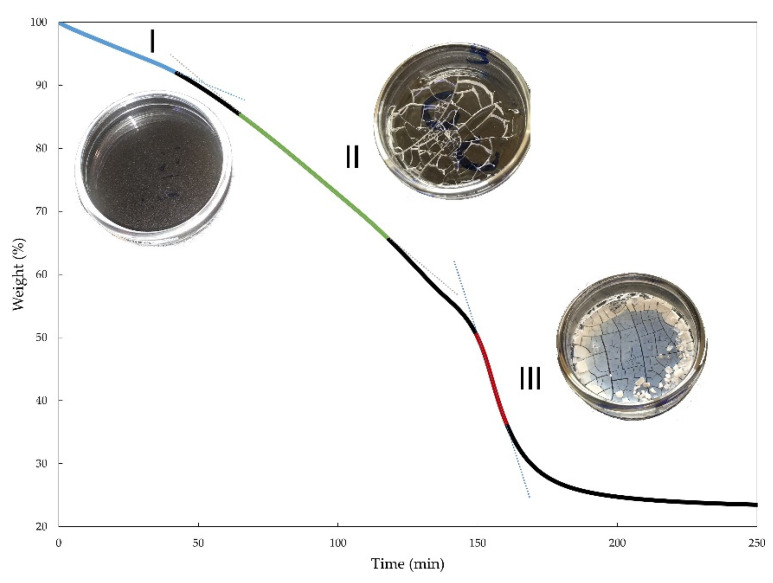
The degradation of STF1000-24 at 140 °C. The degradation steps are highlighted with different colors: I—first step; II—second step, III—third step.

**Figure 5 materials-15-05818-f005:**
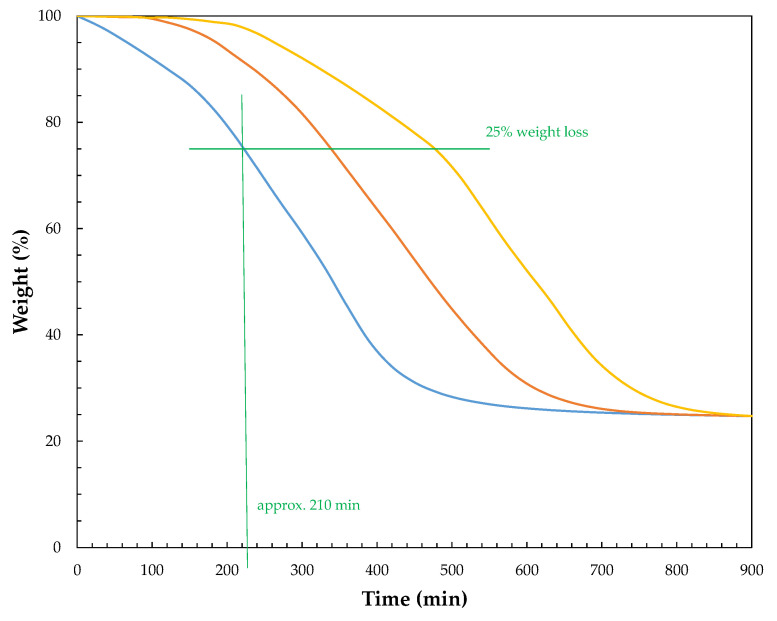
Control lines of weight loss during thermal treatment of STFs at various temperatures: 100 °C (yellow line); 130 °C (orange line); 140 °C (blue line) for the STF1000-24.

**Figure 6 materials-15-05818-f006:**
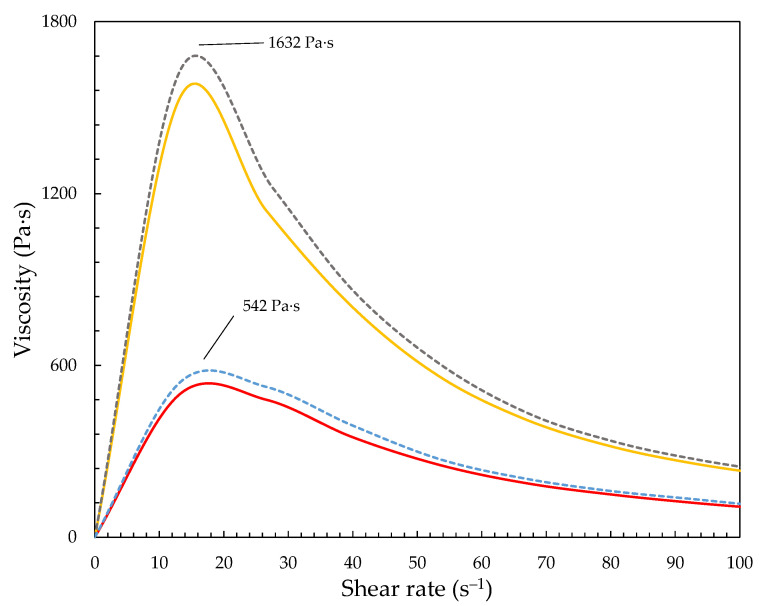
Viscosity vs. shear rate for STFs prior and after thermal treatment: yellow line—STF1000-24; gray dashed line—STF1000-18 after thermal treatment; red line—STF400-24; blue dashed line—STF400-18 after thermal treatment.

**Table 1 materials-15-05818-t001:** Properties of materials used for STF development ^1^.

Abbreviation	Parameter	Value
PPG400	*M*_n_ (g mol^−1^)	~400
	Density (g mL)	1.01
PPG1000	*M*_n_ (g mol^−1^)	~1000
	Density (g cm^−3^)	1.005
	Dynamic viscosity (mPa·s)	78.34
Aerosil^®^200	Specific surface BET (m^2^·g^−1^)	200 ± 25

^1^ According to the Safety Data Sheet provided by the suppliers.

**Table 2 materials-15-05818-t002:** Composition of STFs.

Abbreviation	Ceramic Powder Content in %	Carrier Fluid
STF400-18	18	PPG400
STF400-24	24	
STF1000-18	18	PPG1000
STF1000-24	24	

**Table 3 materials-15-05818-t003:** Weight loss rate (% per min) and maximum heat effect in step III (W·g^–1^) after TG curves at various temperatures.

Step	Temperature/°C				
	100	110	120	130	140
**I**	0.02	0.03	0.05	0.10	0.18
**II**	0.05	0.09	0.12	0.23	0.37
**III**	0.05	0.22	0.28	0.56	1.31
**Max. heat effect**	0.15	0.23	0.35	0.68	1.31

## Data Availability

The data presented in this study are available on reasonable request from the corresponding author.

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
