# Peer review of "Unexpected Method of High-Viscosity Shear Thickening Fluids Based on Polypropylene Glycols Development via Thermal Treatment"

_materials, 2022, doi:10.3390/ma15175818_

Round 1
Reviewer 1 Report
This study analyzed the influence of thermal treatment of shear thickening fluids on their viscosity. viscosity‒shear rate dependence of the STFs, and exhibited a dilatant effect with a maximum viscosity of 1539 Pa·s. The paper may be accepted for publication after the following comments are addressed.
1. the STFs based on PPG400 exhibited approx. three times lower of maximum viscosity values than that of STFs based on PPG1000, more experiments should be performed, for instance, STF based on PPG2000 should be added into comparison.
2. The thermal treatment at 140 degree makes the degradation of the poly(propylene glycol), further makes the increase of maximum viscosity. The mechanism between carrier fluid degradation and viscosity should be discussed in more details.
3. the mechanisms behind the shear thickening behavior are not discussed, molecular chain motion?
Reviewer 2 Report
I regret to inform that I cannot accept the manuscript for publication in Materials. Although the scientific details are good, the work is poorly organized.
A total of 4 materials STF400-18, STF400-24, STF1000-18 and STF1000-24 are studied in this work. TG experiments are conducted for only one of them. But thermal treatments are conducted for all of them(?).
TG experiments are conducted at temperatures between 100 ℃ and 140 ℃, however, thermal treatment results show only 3 temperatures 100 ℃, 130 ℃ and 140 ℃.
Apparently, the authors have discussed the results of thermal treatment of only one material (STF1000-24) but the viscosity after thermal treatments are reported for all the materials(!).
The thermal treatment was conducted for 3 temperatures (which should have been 5 temperatures between 100 ℃ and 140 ℃), but the results are reported for only one temperature (140 ℃ for 210 minutes).
There is no justification on why 210 minutes were selected. If it is considered from the 25% mass loss curve, why the experiments are not repeated at 130 ℃ for 350 min and 100 ℃ for 510 min? The results provided are indeed ‘unexpected’ and not repetitive.
I recommend the authors report the results consistently for all the experiments conducted for all the materials at different temperatures. The work presented now is only a technical report, without validation of the results.
At this moment, the manuscript cannot be accepted.
Some other comments can be found below.
Lines 21 & 22 are unclear. Please refine the whole abstract to provide a clearer description of the work.
Lines 56 & 57 – The sentence provided is very vague. Please give a detailed description.
Why Figure 4 contains only the TG experiments conducted at 140 ℃? I recommend the authors provide another figure with the TG curves at all temperatures, to support their results in Table 2 (which I suppose should be Table 3).
Besides, why the TG experiments are conducted only for STF1000-24? Why not also for STF1000-18? The authors are suggested to give the reasons at the beginning of Section 3.2.
The figure numbers and table numbers are confusing, making it hard to read and understand the results. For example, there are two Figure 4s and two Table 2s.
Why the weight loss during heat treatment of other STFs, STF400-18, STF400-24 and STF1000-18 are not provided?
Round 2
Reviewer 1 Report
The paper can be accepted now